# Fibrosis-4 and 3 indices are independently associated with annual hemoglobin decline in individuals with metabolic disorders: A hospital-based retrospective cohort study

Taiki Hori[1,2,3], Ken-ichi Aihara[4], Yutaka Kawano[4], Takeshi Watanabe[5], Yousuke Kaneko[1,2], Saki Kawata[1,2], Kazuma Shimizu[2], Minae Hosoki[1,2], Kensuke Mori[1,2], Toshiki Otoda[3,4,6], Tomoyuki Yuasa[4], Ken-ichi Matsuoka[3], Shingen Nakamura[4]*

1 Department of Internal Medicine, Anan Medical Center, Tokushima, Japan, 2 Department of Internal Medicine, Tokushima Prefectural Kaifu Hospital, Tokushima, Japan, 3 Department of Hematology, Endocrinology and Metabolism, Tokushima University Graduate School of Biomedical Sciences, Tokushima, Japan, 4 Department of Community Medicine and Medical Science, Tokushima University Graduate School of Biomedical Sciences, Tokushima, Japan, 5 Department of Preventive Medicine, Tokushima University Graduate School of Biomedical Sciences, Tokushima, Japan, 6 Division of General Medicine, Department of Internal Medicine, Nihon University School of Medicine, Tokyo, Japan

* shingen@tokushima-u.ac.jp

## Abstract

The epidemiology of longitudinal changes in hemoglobin (Hb) levels and their associated factors remain unclear. We analyzed data from individuals with metabolic disorders who had been visiting the Department of Internal Medicine at the Anan Medical Center for > 4 years as of December 2021. The relationships between the annual changes in Hb levels (g/dL/year) and clinical confounding factors were also evaluated. Among 481 patients (median age = 66 years), the mean annual change in Hb level was −0.03 ± 0.26 g/dL/year. Multiple regression analysis showed that male sex and sodium-glucose co-transporter-2 inhibitor usage were positively correlated with the annual change in Hb level, whereas the baseline fibrosis-4 (FIB-4) index and fibrosis-3 (FIB-3) indices were associated with the annual decrease in Hb, even after adjusting for age, creatinine level (Cr), and albumin level. Standardized partial regression coefficient (β) of FIB-4 (−0.042, 95% confidence interval [CI]: −0.071, −0.029) and FIB-3 (−0.042, 95% CI: −0.071, −0.013) were comparable to the β-value of Cr- (0.047, 95% CI: −0.078, −0.016). This study is the first to demonstrate that FIB-4 index and FIB-3 index are independently associated with long-term Hb decline in individuals with metabolic disorders. These findings suggest a potential association between liver fibrosis and Hb decline, highlighting the importance of liver health assessment in managing anemia risk.

which permits unrestricted use, distribution, and reproduction in any medium, provided the original author and source are credited.

**Data availability statement:** All relevant data are within the paper and its Supporting Information files.

**Funding:** The author(s) received no specific funding for this work.

**Competing interests:** The authors have declared that no competing interests exist.

## Introduction

Anemia is associated with reduced muscle strength, fatigue, falls, functional decline, cognitive impairment, and depression, which contribute to increased hospitalization, morbidity, and mortality among affected patients [1–3]. The prevalence of anemia increases with age, especially after the age of 50, with incidence rates of 16.5% and 13.0% reported in older men and women, respectively [4]. In a 14-year hospital-based cohort study in the USA, hemoglobin (Hb) changes were associated with prognosis following diagnosis with primary lung, breast, colorectal, or liver cancer [5]. Moreover, a growing body of evidence suggests the association between the decline in Hb level and increased risks of all-cause mortality and diseases such as cardiovascular diseases [6,7]. Therefore, understanding the epidemiology of long-term longitudinal changes in Hb levels may lead to the prevention of anemia and various diseases. However, the longitudinal changes in Hb levels that lead to anemia remain poorly understood.

Hb levels may be associated with liver dysfunction (e.g., hepatitis, fibrosis and cirrhosis) [8]. The fibrosis-4 (FIB-4) index has been clinically used for predicting and monitoring liver fibrosis [9]. It is a non-invasive index calculated using the formula (age [years] × aspartate aminotransferase [AST] [U/L])/ (platelet [Plt] [$10^9$/μL] × √alanine aminotransferase [ALT] [U/L]), developed for predicting liver fibrosis [10]. The FIB-4 index has been proven to effectively stratify patient risk, showing performance comparable to or exceeding that of a liver biopsy, according to a direct comparison [9]. The fibrosis-3 (FIB-3) index can predicts liver fibrosis without age, and is calculated using the following formula (5 × ln AST [IU/L] − 2 × ln ALT [IU/L] − 0.18 × Plt [$10^4$/μL] − 5) [11]. The FIB-3 index has higher accuracy for predicting liver fibrosis caused by metabolic dysfunction-associated steatotic liver disease than the FIB-4 index in patients aged ≥60 years [12].

Individuals with metabolic disorders, such as hypertension, dyslipidemia, and diabetes mellitus, usually require long-term hospital care. In these individuals, anemia is a common complication during the long course of treatment. Anemia is believed to be caused by a combination of erythropoietic stress induced by elevated advanced glycation end products, oxidative stress, endothelial dysfunction, decreased levels of erythropoietin (EPO) due to chronic kidney disease, and medications such as metformin, angiotensin receptor blockers (ARBs), and angiotensin-converting enzyme inhibitors (ACEis) [13,14]. This study aimed to clarify the epidemiology of long-term changes in Hb levels to better understand the process of anemia development and help identify high-risk individuals prone to anemia. We also sought to investigate the associated factors of the annual changes in Hb levels, which may be a new health indicator in outpatients with metabolic disorders.

## Materials and methods

### Study design, participants, and ethics approval statement

In this retrospective cohort study, we recruited outpatients with one or more of the following conditions: hypertension, dyslipidemia, and diabetes mellitus who had been

visiting the Department of Internal Medicine at Anan Medical Center (Anan, Tokushima, Japan) for >4 years as of December 2021. The exclusion criteria were as follows: 1) mean corpuscular volume (MCV) < 80 fL (suspected iron deficiency) or 110 fL< MCV (suspected megaloblastic anemia); 2) treatment with red blood cell transfusion and hematinic drugs, including iron, folic acid, mecobalamin, EPO-stimulating agents, and hypoxia-inducible factor prolyl-hydroxylase inhibitor; 3) advanced cancer, including hematological neoplasms; 4) malnutrition, specifically in cases of liver cirrhosis and 5) bleeding during the observation period

Specifically, the most recent Hb level as of December 2021 was compared with its level recorded ≥ 4 years earlier. The difference was divided by the number of days between measurements and multiplied by 365 to calculate the annual change in Hb levels (annual change in Hb level [g/dL/year] = change in Hb level [g/dL] × 365/ observation period [days]). We examined the associations between baseline blood test results, including the FIB-4 or FIB-3 index, and annual changes in Hb levels. The medications used were defined as those that the patient was taking orally at the end of the observation period, and that had been used for ≥ 2 months.

These data were accessed on February 1, 2022. The authors had access to information that could identify individual participants during data collection, and the information was anonymized after data collection. The requirement for informed consent from each patient was waived in this study. According to the Japanese national guideline of clinical research,"Ethical Guidelines for Medical and Biological Research Involving Human Subjects," the procedures for informed consent may be simplified or waived by taking appropriate measures if some requirements are met [15]. Our research met the criteria and was approved by the Institutional Review Board of Anan Medical Center (approval ID: 202115) in compliance with the national and institutional guidelines. The ethics committee also authorized the conduct of this study using an opt-out system. The study was conducted in accordance with the principles of the Declaration of Helsinki.

## Statistical analysis

Normally distributed continuous variables were expressed as means ± standard deviations, whereas non-normally distributed continuous variables were presented as medians with interquartile ranges (first quartile [Q1]–third quartile [Q3]). Normality was tested using Shapiro–Wilk test. Categorical variables were analyzed using the Fisher's exact test. Numeric variables were assessed using the Mann–Whitney U test, while plot of values that clearly showed normality were assessed using the t-test.

Simple linear regression analysis was employed to evaluate the correlation between baseline Hb levels and the FIB-4 or the FIB-3 index, as well as the association between the annual changes in Hb levels and each variable, including sex, age, general blood test parameters, serum lipid parameters, glycated hemoglobin, FIB-4 index, FIB-3 index, comorbidities, and medication use. Significant variables from these analyses were further examined using multiple linear regression, with multicollinearity assessed accordingly. To calculate standardized regression coefficients (β), all variables were standardized (mean 0, standard deviation 1) prior to linear regression analysis. All statistical analyses were performed using GraphPad Prism Version 9.5.1 (528) for macOS (GraphPad Software, San Diego, CA, USA) and Excel Version 16.100 (Microsoft Office Excel 2025; Microsoft, Richmond, CA, USA). Statistical significance was set at $p < 0.05$.

## Results

### Patient characteristics

There were 631 individuals with metabolic disorders who had been visiting the our hospital for > 4 years as of December 2021. Of these, 481 individuals who did not meet the exclusion criteria. Table 1 presents the demographic characteristics, baseline laboratory data, and medication use. The following key findings were observed: age, observation period, Hb level, creatinine (Cr) level, FIB-4 index, and FIB-3 index of 66 (58−73) years, 1748 (1687−1772) days, 13.9 (13.0–14.9) g/dL, 0.67 (0.55–0.82) mg/dL, 1.38 (1.01–1.83), and 0.15 (−0.78–1.21), respectively. White blood cell (WBC) count, red

**Table 1. Patient characteristics.**

| | Total Participants | Males | Females | *p*-value |
|---|---|---|---|---|
| Number of participants | 481 | 255 | 226 | |
| Age (years)[a] | 66 (58–73) | 66 (59–73) | 66 (57–73) | 0.905 |
| Observation period (days)[a] | 1748 (1687–1772) | 1749 (1689–1774) | 1747 (1682–1768) | 0.195 |
| WBC (×10³/μL)[a] | 5.84 (4.81–7.10) | 6.18 (5.12–7.31) | 5.55 (4.64–6.55) | <0.001 |
| RBC (×10⁶/μL)[a] | 4.55 (4.25-4.90) | 4.68 (4.34-5.08) | 4.46 (4.19-4.68) | <0.001 |
| Hb (g/dL)[a] | 13.9 (13.0-14.9) | 14.6 (13.7-15.6) | 13.4 (12.6-14.0) | <0.001 |
| Hct (%)[a] | 41.4 (39.1-44.3) | 43.0 (40.6-45.9) | 40.2 (38.1-42.2) | <0.001 |
| MCV (fL)[a] | 91.2 (88.6–93.5) | 91.8 (89.3–94.9) | 90.3 (87.8–92.7) | <0.001 |
| Plt (×10³/μL)[a] | 221 (188–255) | 212 (181–247) | 231 (197–264) | <0.001 |
| Alb (g/dL)[a] | 4.3 (4.0–4.4) | 4.3 (4.0–4.5) | 4.3 (4.1–4.4) | 0.498 |
| T-bil (mg/dL)[a] | 0.6 (0.5–0.8) | 0.7 (0.5–0.8) | 0.6 (0.5–0.8) | 0.385 |
| AST (U/L)[a] | 21 (18–26) | 21 (18–27) | 21 (18–24) | 0.306 |
| ALT (U/L)[a] | 21 (16–29) | 22 (17–31) | 20 (15–25) | <0.001 |
| HDL-C (mg/dL)[a] | 51 (43–61) | 48 (40–57) | 55 (47–65) | <0.001 |
| LDL-C (mg/dL)[a] | 108 (90–128) | 109 (88–125) | 106 (92–130) | 0.432 |
| TG (mg/dL)[a] | 120 (85–160) | 125 (90–166) | 120 (80–155) | 0.347 |
| Cr (mg/dL)[a] | 0.67 (0.55–0.82) | 0.79 (0.68–0.92) | 0.55 (0.49–0.65) | <0.001 |
| eGFR (mL/min/1.73 m²)[a] | 80.3 (67.0–93.4) | 76.9 (64.8–89.9) | 83.2 (70.3–96.4) | 0.003 |
| HbA1c (%)[a] | 6.6 (6.1–7.5) | 6.6 (6.1–7.5) | 6.7 (6.2–7.5) | 0.415 |
| FIB-4 index[a] | 1.38 (1.01–1.83) | 1.41 (1.03–1.86) | 1.37 (1.00–1.79) | 0.409 |
| FIB-3 index[a] | 0.15 (−0.78–1.21) | 0.16 (−0.75–1.25) | 0.15 (−0.82–1.14) | 0.155 |
| Hypertension[b] | 334 (69.4) | 185 (72.5) | 149 (65.9) | 0.116 |
| Diabetes mellitus[b] | 353 (73.4) | 206 (80.8) | 147 (65.0) | <0.001 |
| Dyslipidemia[b] | 317 (65.9) | 145 (56.9) | 172 (76.1) | <0.001 |
| ARB or ACEi[b] | 216 (44.9) | 113 (44.3) | 103 (45.6) | 0.781 |
| CCB[b] | 239 (49.7) | 127 (49.8) | 112 (49.6) | 0.957 |
| β-blocker[b] | 29 (6.0) | 16 (6.3) | 13 (5.8) | 0.810 |
| MR antagonist[b] | 10 (2.1) | 3 (1.2) | 7 (3.1) | 0.141 |
| Statin[b] | 242 (50.3) | 112 (43.9) | 130 (57.5) | 0.003 |
| Ezetimibe[b] | 41 (8.5) | 15 (5.9) | 26 (11.5) | 0.028 |
| Other lipid-lowering drugs[b] | 39 (8.1) | 21 (8.2) | 18 (8.0) | 0.914 |
| Insulin[b] | 73 (15.2) | 42 (16.5) | 31 (13.7) | 0.401 |
| SU or Glinide[b] | 103 (21.4) | 64 (25.1) | 39 (17.3) | 0.036 |
| GLP-1 RA[b] | 41 (8.5) | 19 (7.5) | 22 (9.7) | 0.371 |
| DPP4i[b] | 225 (46.8) | 128 (50.2) | 97 (42.9) | 0.110 |
| Metformin[b] | 153 (31.8) | 87 (34.1) | 66 (29.2) | 0.248 |
| αGI[b] | 62 (12.9) | 44 (17.3) | 18 (8.0) | 0.002 |
| Pioglitazone[b] | 7 (1.5) | 4 (1.6) | 3 (1.3) | 0.826 |
| SGLT2i[b] | 129 (26.8) | 82 (32.2) | 47 (20.8) | 0.005 |
| Antiplatelets[b] | 55 (11.4) | 37 (14.5) | 18 (8.0) | 0.024 |
| Anticoagulants[b] | 16 (3.3) | 14 (5.5) | 2 (0.8) | 0.005 |

Mann–Whitney U test for continuous variables. Fisher's exact test for categorical variables. a) median (25%−75%), b) number of participants (%). Abbreviations: WBC: white blood cells; RBC: red blood cells; Hb: hemoglobin; Hct: hematocrit; MCV: mean corpuscular volume; Plt: platelets; Alb: albumin; T-bil: total bilirubin; AST: aspartate aminotransferase; ALT: alanine aminotransferase; HDL-C: high-density lipoprotein cholesterol; LDL-C: low-density lipoprotein cholesterol; TG: triglyceride; Cr: creatinine; eGFR: estimated glomerular filtration rate; FIB-4 index: fibrosis-4 index; FIB-3 index: fibrosis-3 index; ARB: angiotensin receptor blocker; ACEi: angiotensin-converting enzyme inhibitor; CCB: calcium channel blocker; MR antagonist: mineralocorticoid receptor antagonist; SU: sulfonylurea; GLP-1 RA: glucagon-like peptide 1 receptor agonists; DPP4i: dipeptidyl peptidase-4 inhibitor; αGI: α-glucosidase inhibitor; SGLT2i: sodium-glucose co-transporter-2 inhibitor

blood cell count, Hb level (males: 14.6 (13.7–15.6) vs females: 13.4 (12.6–14.0); $p < 0.001$), hematocrit (Hct), MCV, ALT level, and Cr level (males: 0.79 [0.68–0.92] vs females: 0.55 [0.49–0.65]; $p < 0.001$) were higher in males than in females. Females had higher Plt count, estimated glomerular filtration rate (eGFR), and high-density lipoprotein cholesterol levels than males. No significant differences were observed in FIB-4 index (males: 1.41 [1.03–1.86] vs females: 1.37 [1.00–1.79]; $p = 0.409$) FIB-3 index (males: 0.16 [−0.75–1.25] vs females: 0.15 [−0.82–1.14]; $p = 0.155$). Males exhibited a higher prevalence of diabetes mellitus and more use of sulfonylurea (SU), α-glucosidase inhibitor (αGI), and sodium-glucose co-transporter-2 inhibitor (SGLT2i) than females. Conversely, dyslipidemia and the use of statins and ezetimibe were significantly more frequent in females than in males. The use of antiplatelet and anticoagulant agents was more prevalent among males than among females (Table 1).

### Annual changes in Hb level among the total, male, and female participants

The annual change in Hb level in the total participants was −0.03 ± 0.26 g/dL/year. No significant difference in annual change in Hb level was observed between male and female participants (male: −0.03 ± 0.29 vs female: −0.02 ± 0.21 g/dL/year; $p = 0.791$). There were large differences among individuals in the annual change of Hb (Fig 1).

### Associations of the baseline FIB-4 index with baseline Hb levels

Simple linear regression analysis revealed a significant inverse association between the baseline FIB-4 index and baseline Hb levels for all participants (partial regression coefficient [B] = −0.300, $p < 0.001$), males (B = −0.374, $p = 0.002$), and females (B = −0.335, $p = 0.002$; Fig 2 [a][b][c]). However, there was no association between the baseline FIB-3 index and baseline Hb levels for all participants, males, and females (Fig 2 [d][e][f]).

### Factors associated with annual changes in Hb levels

Simple linear regression analysis showed that the baseline FIB-4 index was significantly associated with the annual declines in Hb levels in all participants (B = −0.075, $p < 0.001$), males (B = −0.080, $p < 0.001$), and females (B = −0.066, $p = 0.002$; Fig 3 [a][b][c]). Moreover, FIB-3 index was significantly associated with the annual declines in Hb levels in all participants (B = −0.029, $p < 0.001$), males (B = −0.033, $p = 0.002$), and females (B = −0.023, $p = 0.007$; Fig 3 [d][e][f]).

As shown in Table 2, key clinical factors associated with annual increases in Hb levels included eGFR and SGLT2i. Plt was positively correlated with annual changes in Hb levels in all participants and males but not in females. Metformin

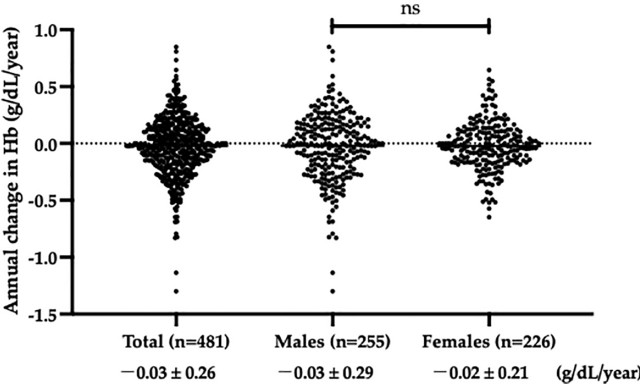

**Fig 1. Comparison of annual changes in hemoglobin levels between males and females.** The annual change in Hb for total participants was −0.03 ± 0.26 g/dL/year. No statistical difference was detected between males and females in the annual change in Hb (males: −0.03 ± 0.29 g/dL/year vs. females: −0.02 ± 0.21 g/dL/year; $p = 0.791$). Abbreviations: Hb: hemoglobin; ns: not significant.

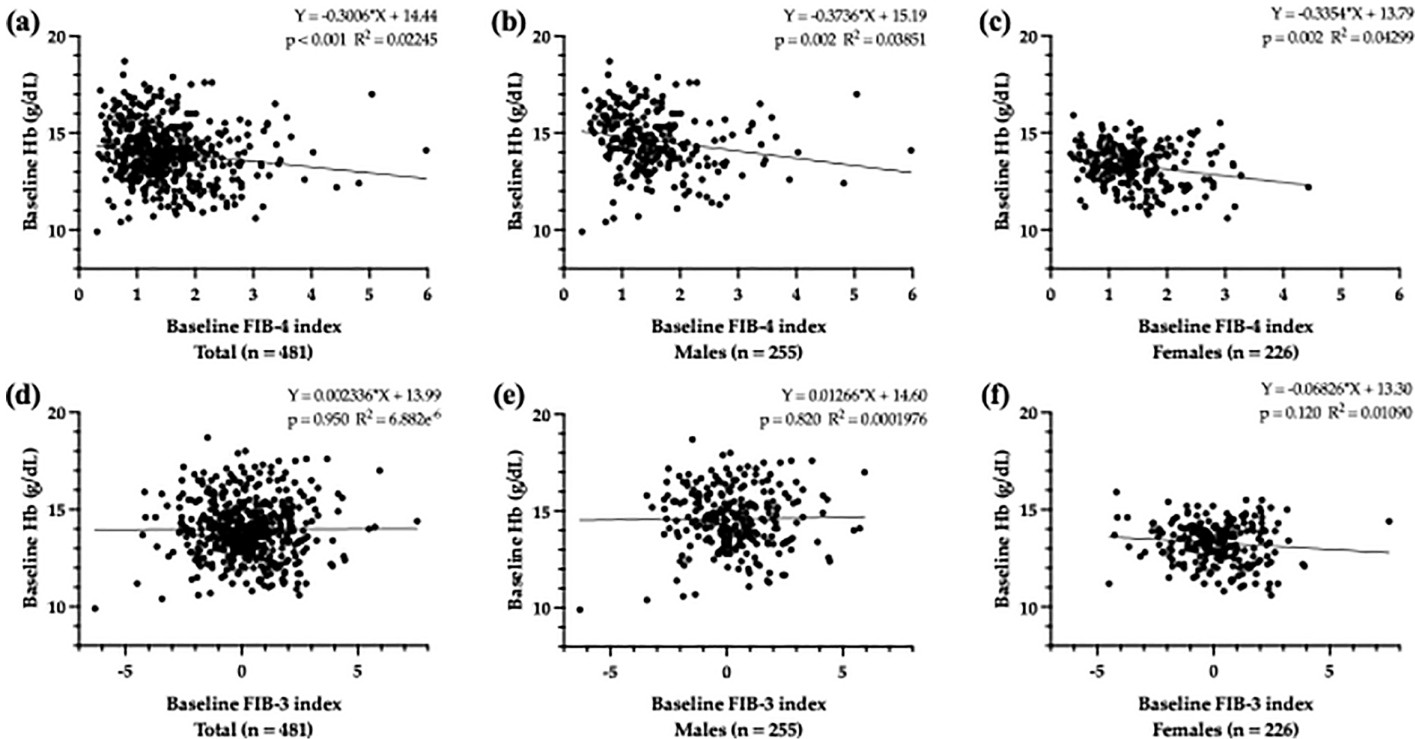

**Fig 2. Association between baseline FIB-4 index or FIB-3 index, and baseline Hb levels.** Scatter plot showing the relationship between baseline FIB-4 index and baseline Hb levels in all participants **(a)**, as well as male (b) and female participants **(c)**. The negative correlations was observed between baseline FIB-4 indices and baseline Hb levels in each group. Scatter plot showing the relationship between baseline FIB-3 index and baseline Hb levels in all participants **(d)**, as well as male (e) and female participants **(f)**. No correlation was observed between baseline FIB-3 indices and baseline Hb levels in each group. Abbreviations: Hb: hemoglobin; FIB-4: fibrosis-4; FIB-3: fibrosis-3.

was positively associated with annual changes in Hb levels in males only. Common clinical factors associated with annual decreases in Hb levels were age, baseline Hb levels, Hct, MCV, albumin (Alb) level, and Cr level. The use of calcium channel blockers and antiplatelets was inversely associated with annual changes in Hb levels in all participants and females but not in males. However, the use of ARBs or ACEis was inversely associated with annual changes in Hb levels in all participants and males but not in females. The use of αGI showed a significantly positive impact on annual changes of Hb levels in females only.

We performed multiple linear regression analyses, incorporating significant variables identified in simple linear regression analysis. As shown in Table 3, male sex and SGLT2i use were significantly associated with annual increases in Hb levels ($p < 0.001$), whereas annual decreases in Hb were independently associated with baseline Hb levels and FIB-4 index ($p < 0.001$). WBC count and Cr level were associated with annual decreases in Hb levels in all participants and males. The use of ARB or ACEi was associated with annual decreases in Hb levels in all participants (Table 3). Standardized partial regression coefficient (β) was calculated to compare the effects of each variable. FIB-4 index was a significant factor influencing changes in Hb (β = −0.042, 95% confidence interval [CI]: −0.071, −0.029), and the β-value was comparable to that of Cr (Fig 4).

Also, multiple linear regression analyses including FIB-3 index was examined. Male sex and SGLT2i use were significantly associated with annual increases in Hb levels ($p < 0.001$), whereas annual decreases in Hb were independently associated with age, baseline Hb levels, and FIB-3 index ($p = 0.004$). WBC count and Cr level were associated with annual

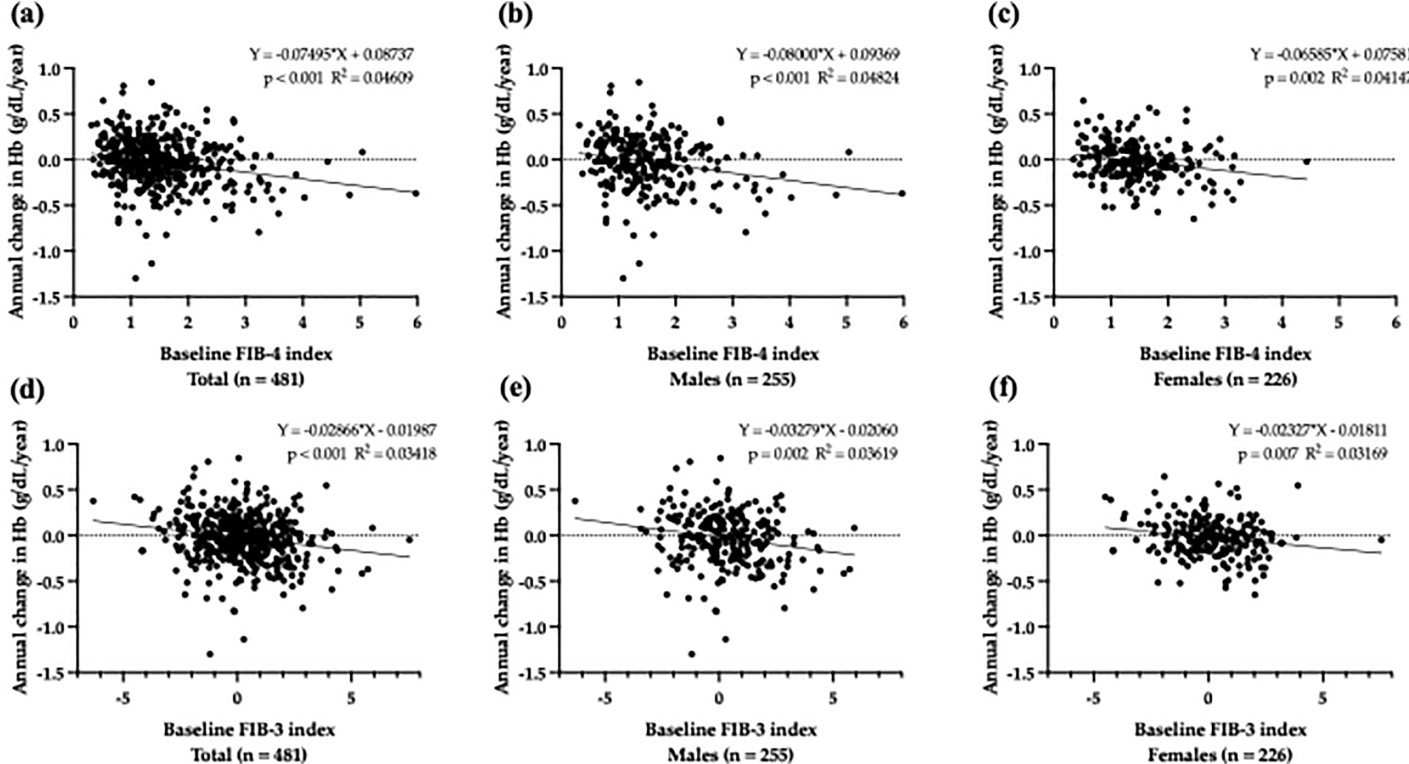

**Fig 3. Association between baseline FIB-4 index or FIB-3 index, and annual changes in Hb levels.** Scatter plot showing the relationship between baseline FIB-4 indices and annual changes in Hb levels in all participants **(a)**, as well as male (b) and female participants **(c)**. The negative correlations was found between baseline FIB-4 index and annual changes in Hb levels in each group. Scatter plot showing the relationship between baseline FIB-3 index and annual changes in Hb levels in all participants **(d)**, as well as male (e) and female participants **(f)**. The negative correlations was found between baseline FIB-3 index and annual changes in Hb levels in each group. Abbreviations: Hb: hemoglobin; FIB-4: fibrosis-4; FIB-3: fibrosis-3.

decreases in Hb levels in all participants and males (Table 4). Standardized partial regression coefficient was calculated to compare the effects of each variable. FIB-3 index was a significant factor influencing changes in Hb ($\beta = -0.042$, 95% CI: $-0.071$, $-0.013$), and the $\beta$-value was comparable to that of age and Cr (Fig 5).

## Discussion

### Annual changes in Hb levels

While short-term studies have examined perioperative Hb changes due to blood transfusions [16], only a few have evaluated changes in Hb levels over time. Zakai et al. reported a median 3-year change in Hb levels of $-0.4$ ($-0.9$ to $0.1$) g/dL (males: $-0.5$ [$-1.1$ to $0.1$] g/dL, females: $-0.5$ [$-1.1$ to $0.1$] g/dL) in 3,758 older healthy adults without anemia [6]. Although their reported annual change in Hb levels appears to be greater than that in this study ($-0.03 \pm 0.26$ g/dL/year), this discrepancy may be partly due to the higher median age of participants in those studies (72.1 years) compared to our mean age of 66 years. A large-scale study examined changes in Hb levels over 2 years in 292,194 healthy individuals aged > 40 years; however, this study only observed increases and decreases in Hb levels but did not quantify the change in Hb levels [7]. Hb decline in individuals without anemia has been associated with an elevated mortality risk (hazard ratio: 1.39–1.60) [6,7], and the annual decrease in Hb levels in these individuals may serve as an important prognostic indicator. In patients with chronic heart failure, patients in Q1 ($-1.64$ [$-6.3$ to $-0.9$] g/dL/year) with changes in Hb divided

**Table 2. Simple linear regression analysis results for determinants of annual changes in Hb levels.**

| Variables | Total participants (n = 481) B (95%CI) | p-value | Males (n = 255) B (95%CI) | p-value | Females (n = 226) B (95%CI) | p-value |
|---|---|---|---|---|---|---|
| Sex (male) | −0.011 (−0.057, 0.036) | 0.656 | | | | |
| Age | −0.004 (−0.006, −0.002) | <0.001 | −0.004 (−0.007, −0.001) | 0.005 | −0.004 (−0.006, −0.001) | 0.002 |
| WBC | −0.003 (−0.015, 0.009) | 0.614 | −0.003 (−0.020, 0.014) | 0.734 | −0.002 (−0.018, 0.014) | 0.794 |
| RBC | −0.045 (−0.094, 0.003) | 0.065 | −0.043 (−0.111, 0.025) | 0.212 | −0.051 (−0.126, 0.024) | 0.179 |
| Hb | −0.033 (−0.049, −0.018) | <0.001 | −0.033 (−0.056, −0.009) | 0.006 | −0.055 (−0.080, −0.030) | <0.001 |
| Hct | −0.012 (−0.018, −0.006) | <0.001 | −0.012 (−0.021, −0.004) | 0.005 | −0.017 (−0.026, −0.008) | <0.001 |
| MCV | −0.012 (−0.017, −0.006) | <0.001 | −0.011 (−0.019, −0.003) | 0.007 | −0.013 (−0.021, −0.006) | <0.001 |
| Plt | 0.001 (0.000, 0.001) | 0.007 | 0.001 (0.000, 0.001) | 0.010 | 0.000 (0.000, 0.001) | 0.366 |
| Alb | −0.134 (−0.215, −0.053) | 0.001 | −0.140 (−0.257, −0.023) | 0.019 | −0.122 (−0.229, −0.016) | 0.025 |
| T-bil | −0.068 (−0.160, 0.024) | 0.147 | −0.054 (−0.200, 0.092) | 0.464 | −0.080 (−0.188, 0.028) | 0.148 |
| AST | −0.002 (−0.003, 0.000) | 0.080 | −0.002 (−0.004, 0.001) | 0.151 | −0.001 (−0.004, 0.002) | 0.423 |
| ALT | 0.000 (−0.001, 0.001) | 0.859 | 0.000 (−0.002, 0.001) | 0.748 | 0.001 (−0.001, 0.003) | 0.548 |
| HDL-C | −0.001 (−0.003, 0.001) | 0.216 | −0.001 (−0.004, 0.002) | 0.501 | −0.002 (−0.004, 0.000) | 0.110 |
| LDL-C | 0.000 (−0.001, 0.001) | 0.894 | −0.001 (−0.002, 0.001) | 0.536 | 0.000 (−0.001, 0.002) | 0.452 |
| TG | −0.134 (0.000, 0.000) | 0.924 | 0.000 (−0.001, 0.001) | 0.960 | 0.000 (−0.001, 0.001) | 0.827 |
| Cr | −0.196 (−0.293, −0.100) | <0.001 | −0.255 (−0.408, −0.102) | 0.001 | −0.218 (−0.384, −0.052) | 0.010 |
| eGFR | 0.003 (0.002, 0.004) | <0.001 | 0.0-03 (0.002, 0.005) | <0.001 | 0.002 (0.001, 0.003) | 0.006 |
| HbA1c | 0.006 (−0.014, 0.026) | 0.559 | 0.007 (−0.022, 0.036) | 0.626 | 0.004 (−0.022, 0.030) | 0.769 |
| FIB-4 index | −0.075 (−0.106, −0.044) | <0.001 | −0.080 (−0.124, −0.036) | <0.001 | −0.066 (−0.108, −0.024) | 0.002 |
| FIB-3 index | −0.029 (−0.042, −0.015) | <0.001 | −0.033 (−0.054, −0.012) | 0.002 | −0.023 (−0.40, −0.006) | 0.007 |
| Hypertension | −0.040 (−0.091, 0.010) | 0.113 | −0.039 (−0.120, 0.043) | 0.349 | −0.041 (−0.099, 0.017) | 0.166 |
| Diabetes Mellitus | 0.055 (0.003, 0.107) | 0.040 | 0.058 (−0.034, 0.150) | 0.214 | 0.059 (0.001, 0.117) | 0.045 |
| Dyslipidemia | 0.035 (−0.014, 0.084) | 0.159 | 0.057 (−0.016, 0.130) | 0.128 | 0.000 (−0.065, 0.065) | 0.996 |
| ARB or ACEi | −0.058 (−0.104, −0.012) | 0.014 | −0.074 (−0.147, −0.002) | 0.045 | −0.040 (−0.095, 0.015) | 0.157 |
| CCB | −0.053 (−0.099, −0.007) | 0.024 | −0.045 (−0.117, 0.028) | 0.225 | −0.063 (−0.118, −0.008) | 0.025 |
| β blocker | −0.011 (−0.109, 0.086) | 0.820 | −0.015 (−0.165, 0.135) | 0.841 | −0.006 (−0.125, 0.113) | 0.923 |
| MR antagonist | 0.084 (−0.078, 0.246) | 0.308 | 0.125 (−0.212, 0.461) | 0.466 | 0.063 (−0.097, 0.223) | 0.436 |
| Statin | 0.044 (−0.002, 0.090) | 0.062 | 0.069 (−0.004, 0.141) | 0.065 | 0.015 (−0.041, 0.071) | 0.607 |
| Ezetimibe | 0.024 (−0.059, 0.107) | 0.570 | 0.141 (−0.013, 0.294) | 0.072 | −0.050 (−0.137, 0.036) | 0.254 |
| Other lipid-lowering drugs | 0.034 (−0.051, 0.119) | 0.428 | 0.100 (−0.031, 0.232) | 0.135 | −0.042 (−0.145, 0.060) | 0.416 |
| Insulin | 0.028 (−0.037, 0.092) | 0.401 | 0.080 (−0.018, 0.177) | 0.108 | −0.040 (−0.120, 0.041) | 0.333 |
| SU or Glinide | 0.021 (−0.035, 0.078) | 0.459 | 0.058 (−0.026, 0.141) | 0.173 | −0.030 (−0.103, 0.044) | 0.426 |
| GLP-1 RA | 0.016 (−0.067, 0.099) | 0.705 | 0.009 (−0.129, 0.147) | 0.897 | 0.021 (−0.073, 0.114) | 0.663 |
| DPP4i | 0.026 (−0.020, 0.072) | 0.262 | 0.044 (−0.028, 0.117) | 0.229 | 0.008 (−0.048, 0.064) | 0.782 |
| Metformin | 0.039 (−0.011, 0.089) | 0.123 | 0.078 (0.002, 0.154) | 0.044 | −0.007 (−0.068, 0.054) | 0.816 |
| αGI | −0.029 (−0.098, 0.040) | 0.408 | 0.019 (−0.077, 0.115) | 0.702 | −0.129 (−0.230, −0.028) | 0.013 |
| Pioglitazone | 0.052 (−0.142, 0.245) | 0.599 | 0.046 (−0.246, 0.338) | 0.756 | 0.060 (−0.182, 0.302) | 0.625 |
| SGLT2i | 0.200 (0.151, 0.249) | <0.001 | 0.195 (0.121, 0.269) | <0.001 | 0.218 (0.156, 0.280) | <0.001 |
| Antiplatelets | −0.084 (−0.156, −0.012) | 0.023 | −0.054 (−0.157, 0.049) | 0.303 | −0.139 (−0.240, −0.038) | 0.007 |
| Anticoagulants | −0.124 (−0.253, 0.004) | 0.058 | −0.119 (−0.277, 0.040) | 0.143 | −0.148 (−0.444, 0.147) | 0.324 |

Abbreviations: B: partial regression coefficient; CI: confidence interval; WBC: white blood cells; RBC: red blood cells; Hb: hemoglobin; Hct: hemato-crit; MCV: mean corpuscular volume; Plt: platelets; Alb: albumin; T-bil: total bilirubin; AST: aspartate aminotransferase; ALT: alanine aminotransferase; HDL-C: high-density lipoprotein cholesterol; LDL-C: low-density lipoprotein cholesterol; TG: triglyceride; Cr: creatinine; eGFR: estimated glomerular filtration rate; FIB-4 index: fibrosis-4 index; FIB-3 index: fibrosis-3 index; ARB: angiotensin receptor blocker; ACEi: angiotensin-converting enzyme inhib-itor; CCB: calcium channel blocker; MR antagonist: mineralocorticoid receptor antagonist; SU: sulfonylurea; GLP-1 RA: glucagon-like peptide 1 receptor agonists; DPP4i: dipeptidyl peptidase-4 inhibitor; αGI: α-glucosidase inhibitor; SGLT2i: sodium-glucose co-transporter-2 inhibitor

Table 3. Multiple linear regression analysis results including FIB-4 index for determinants of annual changes in Hb levels.

| variables | Total participants (n=481) | | | | Males (n=255) | | | | Females (n=226) | | | |
|---|---|---|---|---|---|---|---|---|---|---|---|---|
| | B (95%CI) | SE | p-value | VIF | B (95%CI) | SE | p-value | VIF | B (95%CI) | SE | p-value | VIF |
| Sex (male) | 0.130 (0.062, 0.198) | 0.035 | <0.001 | 1.949 | | | | | | | | |
| Age | −0.002 (−0.005, 0.001) | 0.002 | 0.213 | 2.131 | −0.002 (−0.007, 0.003) | 0.003 | 0.445 | 2.244 | −0.002 (−0.005, 0.002) | 0.002 | 0.371 | 2.235 |
| WBC | −0.017 (−0.031, −0.003) | 0.007 | 0.018 | 1.248 | −0.021 (−0.042, 0.000) | 0.011 | 0.046 | 1.313 | −0.013 (−0.032, 0.006) | 0.010 | 0.175 | 1.210 |
| Hb | −0.062 (−0.083, −0.040) | 0.011 | <0.001 | 1.801 | −0.054 (−0.085, −0.023) | 0.016 | 0.001 | 1.493 | −0.082 (−0.115, −0.050) | 0.016 | <0.001 | 1.569 |
| MCV | −0.005 (−0.012, −0.001) | 0.003 | 0.098 | 1.224 | −0.005 (−0.015, 0.005) | 0.005 | 0.306 | 1.302 | −0.006 (−0.014, 0.001) | 0.004 | 0.108 | 1.191 |
| Alb | −0.075 (−0.153, −0.002) | 0.039 | 0.058 | 1.298 | −0.084 (−0.203, 0.034) | 0.060 | 0.161 | 1.351 | −0.043 (−0.144, 0.058) | 0.051 | 0.404 | 1.405 |
| Cr | −0.195 (−0.323, −0.067) | 0.065 | 0.003 | 1.641 | −0.221 (−0.404, −0.039) | 0.093 | 0.018 | 1.235 | −0.147 (−0.339, −0.045) | 0.097 | 0.132 | 1.370 |
| FIB-4 index | −0.084 (−0.129, −0.039) | 0.023 | <0.001 | 1.907 | −0.093 (−0.159, −0.028) | 0.033 | 0.006 | 1.848 | −0.075 (−0.139, −0.011) | 0.033 | 0.023 | 2.325 |
| ARB or ACEi | −0.056 (−0.111, −0.001) | 0.028 | 0.045 | 1.257 | −0.072 (−0.159, 0.016) | 0.044 | 0.109 | 1.222 | −0.034 (−0.099, 0.031) | 0.033 | 0.306 | 1.411 |
| CCB | 0.000 (−0.055, −0.055) | 0.028 | 0.994 | 1.257 | 0.029 (−0.058, 0.117) | 0.044 | 0.507 | 1.216 | −0.047 (−0.113, 0.020) | 0.034 | 0.166 | 1.476 |
| Statin | 0.021 (−0.030, 0.072) | 0.026 | 0.423 | 1.108 | 0.011 (−0.075, −0.098) | 0.044 | 0.797 | 1.190 | 0.025 (−0.032, 0.083) | 0.029 | 0.385 | 1.086 |
| Metformin | −0.028 (−0.087, 0.030) | 0.030 | 0.339 | 1.198 | −0.004 (−0.097, 0.090) | 0.047 | 0.940 | 1.194 | −0.055 (−0.126, 0.015) | 0.036 | 0.125 | 1.350 |
| αGI | −0.007 (−0.082, 0.068) | 0.038 | 0.859 | 1.106 | −0.008 (−0.117, 0.101) | 0.055 | 0.880 | 1.096 | 0.017 (−0.091, 0.125) | 0.054 | 0.757 | 1.269 |
| SGLT2i | 0.209 (0.149, 0.268) | 0.030 | <0.001 | 1.190 | 0.223 (0.128, 0.318) | 0.048 | <0.001 | 1.262 | 0.199 (0.127, 0.271) | 0.036 | <0.001 | 1.180 |
| Antiplatelets | −0.037 (−0.116, 0.041) | 0.040 | 0.349 | 1.052 | −0.043 (−0.165, 0.078) | 0.061 | 0.483 | 1.074 | −0.038 (−0.135, 0.060) | 0.049 | 0.447 | 1.101 |

Abbreviations: B: partial regression coefficient; CI: confidence interval; SE: standard error; VIF: variance inflation factor; WBC: white blood cells; Hb: hemoglobin; MCV: mean corpuscular volume; Alb: albumin; Cr: creatinine; FIB-4 index: fibrosis-4 index; ARB: angiotensin receptor blocker; ACEi: angiotensin-converting enzyme inhibitor; CCB: calcium channel blocker; αGI: α-glucosidase inhibitor; SGLT2i: sodium-glucose co-transporter-2 inhibitor

into quartile had a greater risk of mortality, hospitalization than patients in Q3 (0.14 [−0.1 to 0.4] g/dL/year) [17]. Notably, this is the first study to reveal the epidemiology of longitudinal changes in Hb levels in individuals with metabolic diseases. Further investigation is needed to determine whether changes in Hb can be used as a broader health indicator in patients other than those with chronic heart failure.

## Factors associated with annual changes in Hb levels

We identified WBC count; baseline Hb and Cr levels; FIB-4 index; FIB-3 index, and the use of ARBs, ACEis, and SGLT2is as significant factors influencing annual changes in Hb levels in metabolic disorders. Age, female sex, diabetes mellitus, and kidney diseases are reportedly associated with Hb decline in older individuals [6]. Additional findings from a study on valsartan in chronic heart failure indicate that Alb, glomerular filtration rate, and diastolic blood pressure positively correlate with increases in Hb levels, whereas C-reactive protein (CRP), left ventricular ejection fraction, B-type natriuretic

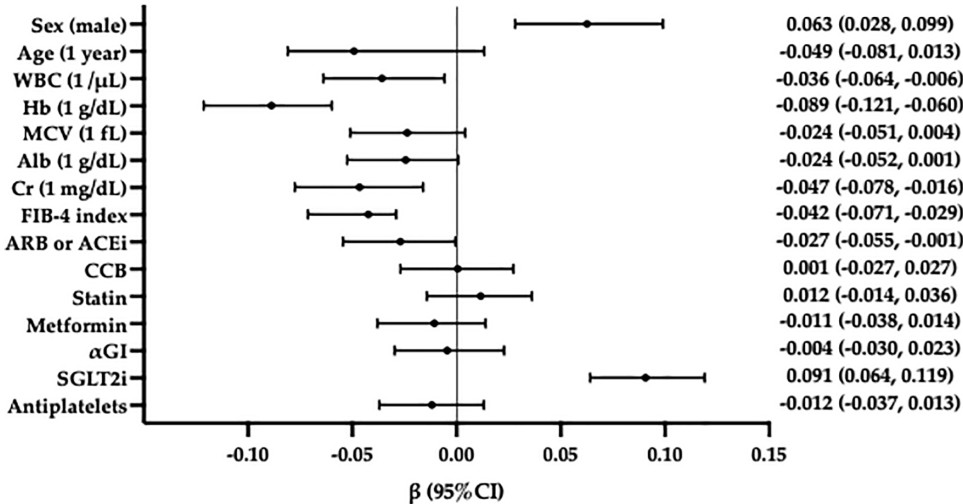

**Fig 4. Standardized partial regression coefficient for annual changes in hemoglobin (Hb) evaluated using multiple linear regression analysis including FIB-4 index.** In all participants, the Standardized partial regression coefficient for the annual change in Hb was evaluated using multiple linear regression analysis. The FIB-4 index had statistical significance independently of other variables, and its Standardized partial regression coefficient (β) was −0.042 (95% CI: −0.071, −0.029) for the annual change in Hb. Among the factors with significant inverse correlation, the β value was similar to that of Cr. Abbreviations: WBC: white blood cells; Hb: hemoglobin; MCV: mean corpuscular volume; Alb: albumin; Cr: creatinine; FIB-4 index: fibrosis-4 index; ARB: angiotensin receptor blocker; ACEi: angiotensin-converting enzyme inhibitor; CCB: calcium channel blocker; αGI: α-glucosidase inhibitor; SGLT2i: sodium-glucose transport protein 2 inhibitors; β: standardized partial regression coefficient; CI: confidence interval.

peptide, and valsartan inversely correlate with Hb level changes over 12 months. This study uniquely establishes the FIB-4 index and FIB-3 index as an independent predictor of annual decline in Hb level.

## Association between the FIB-4 or FIB-3 index, and annual changes in Hb levels

Anemia prevalence is high in patients with advanced liver disease owing to hypersplenism and gastrointestinal bleeding induced by portal hypertension, coagulation dysfunction, and thrombocytopenia [18]. In this study, only 37 (7.7%) patients had an FIB-4 index of ≥2.67, indicative of severe liver fibrosis, whereas liver fibrosis was not considered to be advanced in 213 (44%) with an FIB-4 index ≤ 1.3. This study identified the FIB-4 and FIB-3 indices, a liver fibrosis indicator, as independent risk factor for annual decreases in Hb levels in individuals with metabolic disorders. The FIB-4 index includes age in its formula, therefore the FIB-3 index and age were examined separately in multiple regression analysis, and similar negative correlations with annual change in Hb were observed.

During long-term follow-up of metabolic diseases, age-related blood abnormalities, such as acquired clonal hematopoiesis of indeterminate potential (CHIP), should not be overlooked. The association between CHIP and chronic liver disease has already been reported [19], CHIP may be involved in both liver fibrosis and Hb changes through the pathology of chronic inflammation. Therefore, future studies should verify the relationship between actual changes in Hb levels and liver fibrosis assessed by ultrasound and biopsy to further elucidate the relationship between anemia and the liver.

## Association between inflammation and annual decreases in Hb levels

Inflammation can induce anemia, known as anemia of chronic disease, by increasing hepcidin production in the liver, which consequently impairs iron utilization and shortens erythrocyte lifespan [20]. WBC count and Alb level are clinically non-specific inflammatory markers. However, WBC counts are affected by bone marrow hematopoietic ability, Alb

**Table 4. Multiple linear regression analysis results including FIB-3 index for determinants of annual changes in Hb levels.**

| variables | Total participants (n = 481) | | | | Males (n = 255) | | | | Females (n = 226) | | | |
|---|---|---|---|---|---|---|---|---|---|---|---|---|
| | B (95%CI) | SE | p-value | VIF | B (95%CI) | SE | p-value | VIF | B (95%CI) | SE | p-value | VIF |
| Sex (male) | 0.125 (0.056, −0.194) | 0.035 | <0.001 | 1.956 | | | | | | | | |
| Age | −0.004 (−0.007, −0.001) | 0.001 | 0.003 | 1.587 | −0.004 (−0.009, 0.000) | 0.002 | 0.046 | 1.731 | −0.003 (−0.006, 0.000) | 0.002 | 0.029 | 1.541 |
| WBC | −0.018 (−0.032, −0.004) | 0.007 | 0.013 | 1.289 | −0.024 (−0.046, −0.002) | 0.011 | 0.031 | 1.372 | −0.014 (−0.033, 0.006) | 0.010 | 0.162 | 1.233 |
| Hb | −0.060 (−0.082, −0.038) | 0.011 | <0.001 | 1.828 | −0.052 (−0.083, −0.021) | 0.016 | 0.001 | 1.531 | −0.079 (−0.111, −0.046) | 0.016 | <0.001 | 1.549 |
| MCV | −0.006 (−0.012, 0.001) | 0.003 | 0.089 | 1.233 | −0.005 (−0.016, 0.005) | 0.005 | 0.285 | 1.315 | −0.006 (−0.014, 0.002) | 0.004 | 0.132 | 1.208 |
| Alb | −0.068 (−0.146, 0.010) | 0.040 | 0.089 | 1.299 | −0.073 (−0.192, 0.047) | 0.061 | 0.232 | 1.353 | −0.044 (−0.145, 0.058) | 0.051 | 0.393 | 1.404 |
| Cr | −0.195 (−0.324, −0.065) | 0.066 | 0.003 | 1.649 | −0.220 (−0.405, −0.036) | 0.094 | 0.020 | 1.236 | −0.149 (−0.342, 0.044) | 0.098 | 0.128 | 1.372 |
| FIB-3 index | −0.025 (−0.043, −0.008) | 0.009 | 0.004 | 1.304 | −0.030 (−0.057, −0.003) | 0.014 | 0.032 | 1.330 | −0.023 (−0.044, −0.002) | 0.011 | 0.036 | 1.374 |
| ARB or ACEi | −0.054 (−0.110, 0.001) | 0.028 | 0.055 | 1.257 | −0.074 (−0.163, 0.014) | 0.045 | 0.100 | 1.222 | −0.028 (−0.093, 0.036) | 0.033 | 0.390 | 1.380 |
| CCB | 0.001 (−0.054, 0.056) | 0.028 | 0.964 | 1.257 | 0.032 (−0.056, 0.120) | 0.045 | 0.481 | 1.217 | −0.045 (−0.112, 0.021) | 0.034 | 0.182 | 1.472 |
| Statin | 0.023 (−0.029, 0.075) | 0.026 | 0.378 | 1.109 | 0.014 (−0.073, 0.101) | 0.044 | 0.753 | 1.189 | 0.027 (−0.031, 0.084) | 0.029 | 0.362 | 1.089 |
| Metformin | −0.023 (−0.082, 0.036) | 0.030 | 0.444 | 1.192 | 0.001 (−0.093, 0.095) | 0.048 | 0.984 | 1.191 | −0.049 (−0.120, 0.022) | 0.036 | 0.173 | 1.346 |
| αGI | −0.013 (−0.089, 0.062) | 0.038 | 0.727 | 1.100 | −0.012 (−0.122, 0.097) | 0.056 | 0.825 | 1.095 | 0.002 (−0.104, 0.108) | 0.054 | 0.973 | 1.226 |
| SGLT2i | 0.205 (0.145, 0.264) | 0.030 | <0.001 | 1.185 | 0.220 (0.124, 0.316) | 0.049 | <0.001 | 1.262 | 0.194 (0.122, 0.266) | 0.036 | <0.001 | 1.171 |
| Antiplatelets | −0.037 (−0.116, 0.042) | 0.040 | 0.358 | 1.052 | −0.040 (−0.163, 0.082) | 0.062 | 0.518 | 1.074 | −0.042 (−0.139, 0.055) | 0.049 | 0.397 | 1.094 |

Abbreviations: B: partial regression coefficient; CI: confidence interval; SE: standard error; VIF: variance inflation factor; WBC: white blood cells; Hb: hemoglobin; MCV: mean corpuscular volume; Alb: albumin; Cr: creatinine; FIB-3 index: fibrosis-3 index; ARB: angiotensin receptor blocker; ACEi: angiotensin-converting enzyme inhibitor; CCB: calcium channel blocker; αGI: α-glucosidase inhibitor; SGLT2i: sodium-glucose co-transporter-2 inhibitor

levels are affected by the condition of the liver, and both are affected by nutritional status. In this study, WBC count was associated with annual declines in Hb levels in all participants and males. While baseline Alb levels were associated with baseline Hb levels ($p < 0.001$), Alb level did not correlate with changes in Hb levels in multiple regression analysis. Although CRP data were unavailable, Alb levels and elevated WBC count may reflect the presence of subclinical inflammation.

## Role of renal dysfunction in annual decline in Hb levels

Renal dysfunction and anemia exhibited a reciprocal relationship. While anemia is involved in renal dysfunction progression, reduced EPO production due to renal dysfunction leads to anemia [21,22]. GFR correlates with changes in Hb levels over 12 months in patients with chronic heart failure [17]. In the present study, Cr levels were associated with annual decreases in Hb levels, which is consistent with previously reported findings.

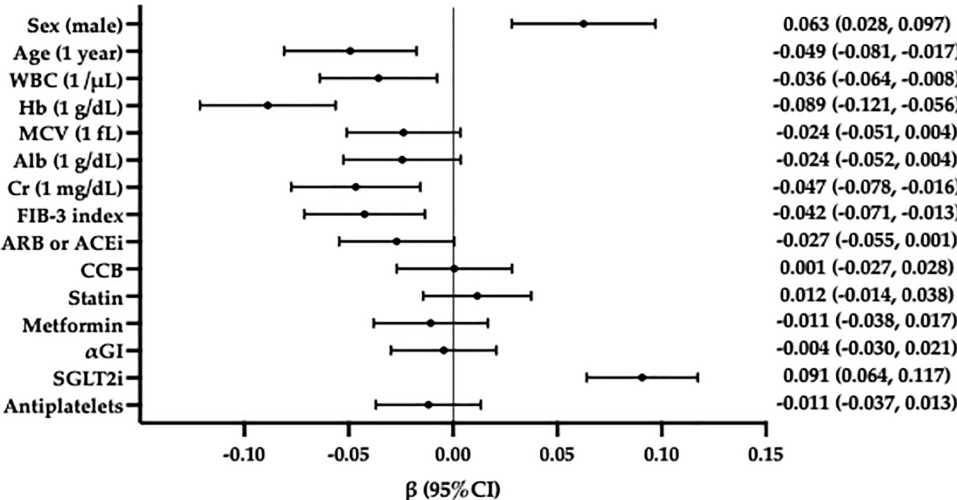

**Fig 5. Standardized partial regression coefficient for annual changes in hemoglobin (Hb) evaluated using multiple linear regression analysis including FIB-3 index.** In all participants, the Standardized partial regression coefficient for the annual change in Hb was evaluated using multiple linear regression analysis. The FIB-3 index had statistical significance independently of other variables, and its Standardized partial regression coefficient (β) was −0.042 (95% CI: −0.071, −0.013) for the annual change in Hb. Among the factors with significant inverse correlation, the β value was similar to that of age and Cr. Abbreviations: WBC: white blood cells; Hb: hemoglobin; MCV: mean corpuscular volume; Alb: albumin; Cr: creatinine; FIB-3 index: fibrosis-3 index; ARB: angiotensin receptor blocker; ACEi: angiotensin-converting enzyme inhibitor; CCB: calcium channel blocker; αGI: α-glucosidase inhibitor; SGLT2i: sodium-glucose transport protein 2 inhibitors; β: standardized partial regression coefficient; CI: confidence interval.

## Association between medication and annual changes in Hb levels

ARBs and ACEis are commonly used to prevent the progression of renal and heart failure. A meta-analysis reported that the use of ARBs or ACEis was associated with anemia [23]. Although the mechanism of anemia is not well-established, ACEis have been reported to decrease plasma EPO concentration, and ARBs have been shown to inhibit angiotensin II-induced proliferation of early erythroid progenitors [24,25]. SGLT2 is increasingly used for renal and cardiovascular protection and glycemic control, and has demonstrated a strong protective effect, partly through anemia correction [26,27]. They increase Hb levels over time by increasing EPO and reducing hepcidin levels [26,28,29]. The present study confirmed significant long-term Hb level changes associated with these agents, aligning with findings from previous research. To our knowledge, it is the first study to evaluate the effect of SGLT2 inhibitors on Hb level changes over 4 years. These factors are likely to be interrelated. Liver fibrosis, inflammation, and renal dysfunction share overlapping pathophysiological pathways [30,31].

## Limitations

This study conducted a before-and-after comparison across two time periods with a long median observation period of 1748 d. Therefore, the effects of medication changes or disease events that occurred during the interval could not be completely eliminated. This study is based on the assumption of outpatients with stable medical conditions, but changes in medication, such as discontinuation or addition of drugs, occur over many years. This study focused on the medications being taken at the end of the study period. Therefore, a limitation is that it does not consider how long the medications were taken or previous medications. Although an overall trend is observed, the exact reasons behind the Hb level changes in individuals remain uncertain, preventing a comprehensive evaluation of Hb decline. Anemia treated with medication or bleeding during the observation periods were excluded; however, subclinical iron deficiency or bleeding may not have been entirely excluded. Iron metabolism including ferritin levels has not been examined, and while we

have used MCV<80fL as an exclusion criterion to eliminate iron deficiency as much as possible, this is a significant limitation.

Liver fibrosis was estimated using the FIB-4 and FIB-3 indices, but without liver histology or ultrasound confirmation. Therefore, further studies are needed to evaluate the relationship between histological changes in the liver and annual changes in Hb levels. However, because of the variability in fibrosis and its invasiveness, we believe that evaluation using these indices is as valid as liver biopsy. This was a retrospective cohort study, and unmeasured confounding factors such as CRP, body mass index, and drinking habits could not be fully controlled. Therefore, caution is required to interpret the causal relationship between liver fibrosis and Hb-change. Prospective research based on the present study is warranted. Furthermore, long-term outcomes regarding Hb level change and actual event occurrence could not be followed up on. Further investigation is required to include these potential confounding factors.

## Conclusions

This study clarified the epidemiology of actual annual changes in Hb levels and their associated factors in individuals with metabolic disorders. The FIB-4 and FIB-3 indices were associated with the annual changes in Hb levels independent of other variables, and the relationship liver fibrosis and progression of anemia was suggested.

## Supporting information

**S1 Data. The dataset used for the analysis.**
(XLSX)

## Acknowledgments

We would like to thank Editage (www.editage.jp) for the English language editing.

## Author contributions

**Conceptualization:** Taiki Hori, Ken-ichi Aihara, Shingen Nakamura.

**Data curation:** Taiki Hori, Ken-ichi Aihara, Yousuke Kaneko, Saki Kawata, Kazuma Shimizu, Minae Hosoki, Kensuke Mori, Toshiki Otoda, Tomoyuki Yuasa, Shingen Nakamura.

**Formal analysis:** Taiki Hori, Ken-ichi Aihara, Yutaka Kawano, Takeshi Watanabe, Shingen Nakamura.

**Project administration:** Shingen Nakamura.

**Writing – original draft:** Taiki Hori.

**Writing – review & editing:** Ken-ichi Aihara, Yutaka Kawano, Takeshi Watanabe, Ken-ichi Matsuoka, Shingen Nakamura.

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
