## [Decision Letter · Decision Letter 0]

1 Oct 2025

Dear Dr.  nakamura,

Thank you for submitting your manuscript to PLOS ONE. After careful consideration, we feel that it has merit but does not fully meet PLOS ONE’s publication criteria as it currently stands. Therefore, we invite you to submit a revised version of the manuscript that addresses the points raised during the review process.

We look forward to receiving your revised manuscript.

Kind regards,

Marwan Salih Al-Nimer, MD, PhD

Academic Editor

PLOS ONE

Journal Requirements:

3. In the online submission form, you indicated that “Ethics Committee (contact via Shingen Nakamura) for researchers who meet the criteria for access to confidential data.”

Additional Editor Comments:

**Major revision**

Methods: Identify in the inclusion criteria the range of MCV value to eliminate the iron, folate, vitamin B1 2 deficiency. Clarify the calculating sample size. Clarify how the annual change of HB was calculated?

Statistical section: Add the test of normality ? is it Shapiro Wilks test? Simple linear regression analysis? do you mean bivariate or multiple variable

Results: multicollinearity testing need to add the Bias +/-SE results.

References: type according to the Journal style

Reviewer's Responses to Questions

**Comments to the Author**

1. Is the manuscript technically sound, and do the data support the conclusions?

Reviewer #1: Partly

Reviewer #2: Partly

2. Has the statistical analysis been performed appropriately and rigorously?

Reviewer #1: Yes

Reviewer #2: Yes

3. Have the authors made all data underlying the findings in their manuscript fully available?

Reviewer #1: Yes

Reviewer #2: Yes

4. Is the manuscript presented in an intelligible fashion and written in standard English?

Reviewer #1: Yes

Reviewer #2: No

Reviewer #1: This paper investigates how markers of liver fibrosis, specifically FIB-4 and FIB-3 indices predict longitudinal declines in hemoglobin levels in people with metabolic disorders based on a hospital-based retrospective cohort study.

Limitations

- Title is too long

- Retrospective Design can not firmly establish causal relationships.

- Fibrosis Assessment: Indices used instead of histological confirmation (biopsy or ultrasound).

- The exclusion of "suspected iron deficiency" based solely on MCV <80 fL is a significant limitation. MCV is a late marker of iron deficiency. Many patients with iron deficiency, especially in the context of chronic disease, can have a normal MCV. The lack of data on ferritin, transferrin saturation, or C-reactive protein (CRP) to rule out iron deficiency or anemia of inflammation is a critical point that weakens the conclusions.

-  It is unclear how medication use was defined. Was it used at baseline, at the end of the study, or continuous use throughout the observation period? Clarify in the Methods section how medication use was defined and accounted for the analysis.

- The results show that both indices perform very similarly. The discussion could better elaborate on why using both indices was necessary and what the comparative findings imply.

Results (Page 5, Line 150): "creatine (Cr) level" should be "creatinine (Cr) level".

Results (Page 11, Line 211): "Cr level were associated with annual decreases" should be "Cr level was associated...".

Reviewer #2: In this paper Hori et al tried to find an association between FIB-4 and FIB-3 indexes and anual decrease in hemoglobin, since lower hemoglobin has been associated with adverse outcomes.

They conducted a retrospective study including patients with metabolic disease who are at risk of liver fibrosis and documented annual change in Hb and its relationship to several other laboratory markers and treamtents.

There are some interesting findings from the study but I have some important observations:

1. In the methods section reasons for excluding patients include anemia and some other conditions that can cause alterations in hemoglobin levels. However, patients with history of gastrointestinal bleeding were not excluded and that might be a cause of acute hemoglobin decline and might not necessarily require transfusion or be diagnosed as iron deficiency. I think you should consider excluding this patients too.

2. There are some mistakes in your grammar, please check it again. The first part of the results section is very confusing so I would suggest rewriting it to make it more clear.

3. Regarding the analyses I understand you did adjustments for variables that resulted significant in the single linear regression, however, this does not necessarily explain biological interactions and confounding. I would suggest you explain better which variables you adjusted for and why. I would highly recommend making a directed acyclic graph for better explanation of causality.

4. In the discussion you explain some biological plausible explanations for the relationship between some variables and lower hemoglobin, but some are connected between them and may need further explanation.

**Do you want your identity to be public for this peer review?** For information about this choice, including consent withdrawal, please see our Privacy Policy

Reviewer #1: **Yes: ** Mohamed Saad Hashim

Reviewer #2: **Yes: ** Mercedes Aguilar Soto

---

## [Author Response · Author response to Decision Letter 1]

19 Nov 2025

Reviewer #1: This paper investigates how markers of liver fibrosis, specifically FIB-4 and FIB-3 indices predict longitudinal declines in hemoglobin levels in people with metabolic disorders based on a hospital-based retrospective cohort study.

→Thank you for your careful review. You also mentioned the limitations of this study, which allowed us to consider the limitations of this research and plans for future research. Your comments are important in helping us to improve our research.

Limitations

Title is too long

Response: Thank you for pointing that out. We have changed the title to “Fibrosis-4 and 3 indices are independently associated with annual hemoglobin decline in individuals with metabolic disorders: A hospital-based retrospective cohort study”. Thanks to your suggestion, the content is understood more concisely by the readers.

Retrospective Design can not firmly establish causal relationships.

Response: Thank you for the valuable suggestion. Taking the reviewer’s comment into account, we have added a description about the limitation “This was a retrospective cohort study, and unmeasured confounding factors such as CRP, BMI, and drinking habits could not be fully controlled. Therefore, caution is required to interpret the causal relationship between liver fibrosis and Hb-change. Prospective research based on the present study is warranted.”

Fibrosis Assessment: Indices used instead of histological confirmation (biopsy or ultrasound).

Response: Thank you for pointing this out. The fact that histological or ultrasound examinations could not be performed is noted as a limitation on page 17, line 353. However, considering the invasiveness, cost, and manpower involved, it is not realistic to perform this test on all patients suspected of very mild fibrosis in clinical practice. The FIB-4/3 index allows for repeated and simple evaluation; it has been shown to correlate with actual liver fibrosis, respectively. When validating this research in prospective studies, it is necessary to include histological or ultrasound evaluations. Thank you for your important opinion.

The exclusion of "suspected iron deficiency" based solely on MCV <80 fL is a significant limitation. MCV is a late marker of iron deficiency. Many patients with iron deficiency, especially in the context of chronic disease, can have a normal MCV. The lack of data on ferritin, transferrin saturation, or C-reactive protein (CRP) to rule out iron deficiency or anemia of inflammation is a critical point that weakens the conclusions.

Response: We thank the reviewer for this comment. As you're pointing out, this is a significant limitation because these markers have not been estimated. It was difficult because this study was a retrospective design. We will further investigate Hb changes and their related factors in a prospective study. This study is a pilot study for future research. We added more details at “limitations” on page 17, line 358.

It is unclear how medication use was defined. Was it used at baseline, at the end of the study, or continuous use throughout the observation period? Clarify in the Methods section how medication use was defined and accounted for the analysis.

Response: Thank you for pointing this out. We defined the medication used as those that the patient was taking orally at the end of the observation period and that had been used for more than 2 months. We added the sentence to the section “Materials and methods - Study design, participants, and ethics approval statement” on page 4, line 108. And, We have added more details at “limitations” on page 16, line 334.

The results show that both indices perform very similarly. The discussion could better elaborate on why using both indices was necessary and what the comparative findings imply.

Response: Thank you for pointing that out. The formula of the FIB-4 index includes age. To avoid confounding effects due to age, we also evaluated the results using the FIB-3 index (age is not included in the calculation), which demonstrated a correlation with Hb changes. We added the sentence the section “Discussion ” on page 15, line 271 in the revised manuscript.

Results (Page 5, Line 150): "creatine (Cr) level" should be "creatinine (Cr) level".

Response: Thank you for your careful review. We have corrected the words you pointed out.

Results (Page 11, Line 211): "Cr level were associated with annual decreases" should be "Cr level was associated...".

Response: Thank you for pointing that out. The sentence is “WBC count and Cr level were associated…” and we have written it in the plural because it has two subjects.

Reviewer #2: In this paper Hori et al tried to find an association between FIB-4 and FIB-3 indexes and anual decrease in hemoglobin, since lower hemoglobin has been associated with adverse outcomes.

They conducted a retrospective study including patients with metabolic disease who are at risk of liver fibrosis and documented annual change in Hb and its relationship to several other laboratory markers and treamtents.

There are some interesting findings from the study but I have some important observations:

→Thank you for your careful review. You mentioned the need to further detail the relationships between the factors involved in Hb changes. Your comments are important in helping us to further improve this research.

1. In the methods section reasons for excluding patients include anemia and some other conditions that can cause alterations in hemoglobin levels. However, patients with history of gastrointestinal bleeding were not excluded and that might be a cause of acute hemoglobin decline and might not necessarily require transfusion or be diagnosed as iron deficiency. I think you should consider excluding this patients too.

Response: Thank you for pointing that out. As you mentioned, gastrointestinal bleeding affects the change in Hb levels. Patients with bleeding that was documented in their medical records were originally excluded, therefore we added to the exclusion criteria on page 4, line 100. However, bleeding episodes not reported by the patient or minor bleeding that did not require clinical treatment may be overlooked. Future research should include regular follow-up of iron dynamics to assess subclinical iron deficiency.

2. There are some mistakes in your grammar, please check it again. The first part of the results section is very confusing so I would suggest rewriting it to make it more clear.

Response: Thank you for reviewing the manuscript, . We reviewed and revised throughout the manuscript.

3. Regarding the analyses I understand you did adjustments for variables that resulted significant in the single linear regression,

however, this does not necessarily explain biological interactions and confounding. I would suggest you explain better which variables you adjusted for and why. I would highly recommend making a directed acyclic graph for a better explanation of causality.

Response: Thank you for the valuable suggestion. We conducted an exploratory analysis to find factors associated with Hb changes in the present study, since past research has not clarified much about the factors related to changes in Hb. As the reviewer suggested, we would like to use a directed acyclic graph to explain which variables we adjusted for and why as you suggested, when we will conduct a prospective study to show the association between factors we found in the present study and Hb changes adjusting for potential confounders.

4. In the discussion you explain some biological plausible explanations for the relationship between some variables and lower hemoglobin, but some are connected between them and may need further explanation.

Response: Thank you for the careful review. Various organs and hormones are involved in the regulation of hemoglobin levels in the body. We examined each of the factors involved in hemoglobin separately, but they are also interconnected. This study did not examine the relationships between those factors. While the underlying causes may be aging or chronic inflammation, or CHIP, these factors were not included in this study. Future research is needed to investigate whether these underlying factors are associated with Hb decline.

---

## [Editor Report · Decision Letter 1]

23 Nov 2025

Fibrosis-4 and 3 indices are independently associated with annual hemoglobin decline in individuals with metabolic disorders: A hospital-based retrospective cohort study

PONE-D-25-49113R1

Dear Dr. Shingen Nakamura,

We’re pleased to inform you that your manuscript has been judged scientifically suitable for publication and will be formally accepted for publication once it meets all outstanding technical requirements.

Kind regards,

Marwan Salih Al-Nimer, MD, PhD

Academic Editor

PLOS ONE

Additional Editor Comments (optional):

No comments
---

## [Editor Report · Acceptance letter]

PONE-D-25-49113R1

PLOS ONE

Dear Dr. Nakamura,

I'm pleased to inform you that your manuscript has been deemed suitable for publication in PLOS ONE. Congratulations! Your manuscript is now being handed over to our production team.

Kind regards,

on behalf of

Professor Marwan Salih Al-Nimer

Academic Editor

PLOS ONE